# XBoundNet++: Interpretable, Uncertainty-Aware Segmentation of Ambiguous CT Labels—Kidney Ablation Zone Study

**Abstract.** Kidney ablation therapy is a minimally invasive procedure used to treat renal tumours. Evaluating treatment success for planning follow-up care relies on accurate kidney ablation zone (KAZ) segmentation in post-operative CT images. However, manual segmentation is time-consuming and prone to inter-observer variability and traditional segmentation is challenging because ground truth labels only provide a partial estimate of the area of interest. This challenge is prevalent in many interventional and surgical contexts, yet understudied in the medical imaging domain. Therefore, segmenting the area of interest requires careful attention to the specific clinical needs of the resulting deep learning framework, including adding model interpretability and uncertainty estimation for further clinical review. We introduce a deep learning framework, XBoundNet++, that permits (1) precise segmentation of the boundary, (2) detailed attention maps for model layer-wise interpretability, and (3) model uncertainty estimation based on Bayesian Monte-Carlo dropouts and model ensembles. The model was trained and evaluated using a nested 5-fold cross-validation on a local dataset of 76 patients (with 912 CT 2D radial slices), collected at London Health Sciences Centre, which included manually annotated KAZs. Quantitative analysis showed that XBoundNet++ achieved promising segmentation results, including 88% precision, 83% recall, 84% DSC, 74% Jaccard, 6.89-pixel Mean Absolute Distance (MAD), -0.60-pixel Mean Signed Distance (MSD), and a 19.86-pixel Hausdorff distance (HD). Furthermore, heatmaps at each layer, probability and uncertainty maps, and uncertainty estimation at several thresholds indicates model trustworthiness, confidence, and justification for predictions.

**Keywords:** Kidney Ablation, Segmentation, Interpretability, Uncertainty

## 1    Introduction

Many segmentation methods perform well for known structures (e.g., kidney, liver in CT) [1, 9], as well as pathological structures (e.g., brain tumours) [10]. However, supervised learning requires the ground truth labels for objects of interest for training, but there are contexts in which ground-truth labels are generally challenging to obtain. Furthermore, it is well-established that only coarse and rough labels are provided by

clinicians with the understanding that these are rough estimates that identify the area of interest. This context is common in clinical interventions, for example, where pathology boundaries are unclear due to poor modality contrast (e.g., CT, MRI, ultrasound). Furthermore, additional challenges are presented in these types of clinical contexts as they typically consist of a limited number of cases. These contexts are not well studied in the literature and require particular care in terms of providing the clinician with model transparency and model uncertainty in order to trust the results and review the areas of relevance. This is crucial to enable trustworthy AI, helping with:

- **Trust**: Clinicians trust in the model will increase if they can understand how it arrived at the prediction, which is important for some clinical applications.
- **Safety**: Evaluating and monitoring prediction uncertainty to prevent a segmentation error and uncertainty from cascading into a clinical error.
- **Transparency**: Detecting biases and model failures and making edits.
- **Improved models**: Gaining insight as to how and what our model learns.

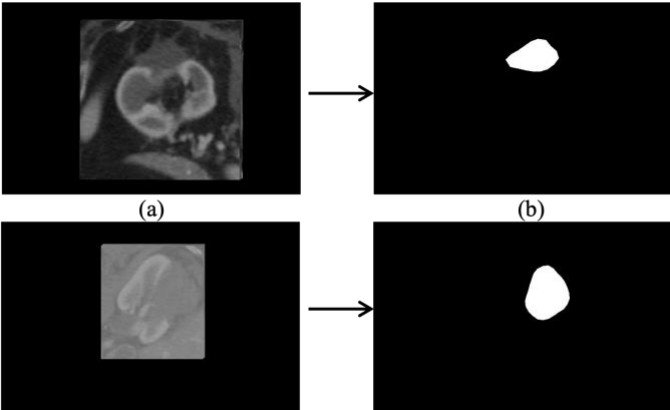

**Fig. 1.** Two sample patient images from the dataset, where (a) are raw images, and (b) clinically annotated images.

In this paper, we consider post-treatment delineation of the ablation zone in kidney CT images. Kidney cancer, or renal cell carcinoma, is one of the most prevalent urological malignancies worldwide. For patients unfit for surgical intervention, thermal ablation therapies like microwave or radiofrequency ablation offer a minimally invasive alternative. These procedures aim to destroy malignant cells by creating a "kidney ablation zone" (KAZ) that encapsulates the tumour and surrounding margin. Post-treatment assessment depends on accurately identifying the entirety of the KAZ in follow-up CT scans, which is a critical task for determining treatment success and guiding subsequent care [4].

In this work, we introduce an *XBoundNet++*, an eXplainable Boundary-Aware modified ResU-Net++, a novel deep learning segmentation framework designed to provide clinicians with high quality segmentation results, model transparency, interpretable tools, and uncertainty estimation using Bayesian Monte-Carlo (MC) dropout [5]. Our framework is aimed at shifting clinical practice from unclear binary

masks to interpretable tools that explicitly provide confidence, uncertainty, probability, and transparency. We created an end-to-end pipeline to preprocess an image (as seen in **Fig. 1**), feed it into our model, generate segmentations of high quality that outperform other state-of-the-art models, provide comprehensive layer-wise transparency, and produce epistemic-uncertainty with probability maps.

## 2      Methods

### 2.1      XBoundNet++ Segmentation Network and Training

We propose XBoundNet++, an ensemble-based four-level modified U-Net [13] in **Fig. 2**, which introduces architectural elements that explicitly promote feature relevance, spatial focus, and post-hoc transparency. Our architecture integrates components from LeXNet++ [3], ResNet [6], attention mechanisms [11], Squeeze & Excitation (SE) [7], STEM [12], and several advanced architectures.

The Atrous Spatial Pooling Pyramid (ASPP) bridge, connecting the encoder and decoder, captured multi-scale context while maintaining dimensionality. It applied convolutions with dilations of 1, 6, 12, and 18 [2], performed a summation to merge the features, and applied a BN and ReLU activation.

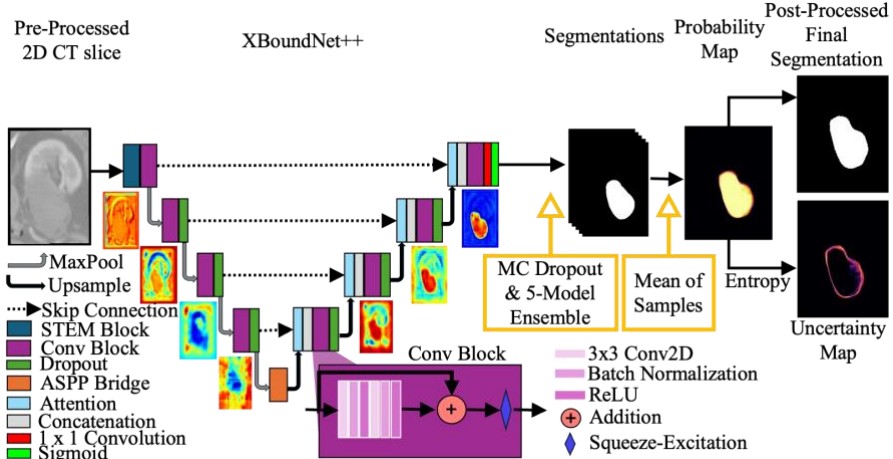

**Fig. 2.** Network pipeline and architecture with layer-wise activation maps.

Attention Gate blocks were introduced to selectively propagate relevant features during upsampling. They compute spatial attention maps via 1×1 convolutions and ReLU-sigmoid activation, suppressing irrelevant activations and enhancing decoder focus on the ablation zone.

The network was optimized with Adam (learning rate $10^{-4}$, batch size 4) and a custom combined loss of Log-Dice ($\alpha = 0.7$) and binary cross-entropy ($\alpha = 0.3$), as Dice addresses class imbalance, while BCE improves per-pixel calibration., giving

probabilistic outputs that can be further used for uncertainty estimation. Early stopping (patience = 50) and Reduce-LR-on-Plateau (factor 0.1, patience = 15) were employed to prevent over-fitting and facilitate convergence. The final combined loss is:

$$CombinedLoss(y, \hat{y}) = \alpha * LogDiceLoss(y, \hat{y}) + (1 - \alpha) * BCE(y, \hat{y}) \qquad (1)$$

where y is the ground truth, ŷ is the prediction, and α is set to 0.7.

For each of the five patient-wise folds, we trained five instances of XBoundNet++ with differing seeds, yielding 25 independent models in total. Altering the seed affects weight initialization, alters the stochastic augmentation stream, and changes the sequence of dropout masks encountered during optimization. The resulting ensemble enhances predictive stability, generalizes the small dataset, and forms the basis for the uncertainty analysis described in the next section.

## 2.2    Layer-wise Heatmap Generation

We propose a custom Gradient-weighted Class Activation Mapping (Grad-CAM) [14] method that helps visualize which regions of an image had the most influence on the model by analyzing gradient-weighted activations that serve as spatial attention maps. First, the image is processed by the model, a class prediction is made, and during backpropagation, the gradients of the prediction are computed for the feature map at the chosen convolutional layer. Finally, the results are passed through a ReLU activation (and upsampled if necessary) to produce the heatmap for any given convolutional layer in the network.

We then extract heatmaps from every convolutional block across the network to observe how feature abstraction evolves at different depths. This strategy allows us to visually trace the information flow and decision-making within the network, revealing where and how the network's focus shifts, from low-level texture extraction to high-level semantic boundary recognition.

## 2.3    Model Inference and Uncertainty Estimation

During training, dropout mitigates over-fitting. During inference, the five seed-specific trained network models retained from each outer fold are evaluated with dropout kept active. For every unseen test slice, we generate 50 stochastic outputs using Bayesian MC dropout, yielding a collection of 250 predictions per slice. We then average this collection to create an ensemble-predictor, producing a probability map $p$ for KAZ segmentation.

We use the probability map to generate an uncertainty map using normalized entropy, $H$, as a measure of uncertainty as portrayed below:

$$H = -[p \log(p) + (1 - p) \log(1 - p)]. \qquad (2)$$

We pool the raw predictions of the validation slices and fit a one-dimensional logistic-regression calibrator. The fitted sigmoid is saved and applied to all test-set probabilities, producing a calibrated map. We then use a 0.4 threshold to binarize the

prediction so that values are either 0 or 1. Next, we perform a morphological closing operation to seal small holes or gaps in the prediction if necessary. We also examined whether numerous disconnected components were present, in which case the largest foreground component is retained, and all other objects are suppressed. This didn't apply to instances where cysts are larger than the KAZ; in such a case, the second largest component is selected.

The cleaned final segmentation mask was then resized to the original CT image size (510 × 788 pixels) with Lanczos-4 interpolation and written to disk as an 8-bit BMP.

### 2.4      Evaluation Metrics

Standard pixel and distance-based metrics were used to assess both technical and clinical segmentation quality. These metrics included the Dice similarity coefficient (DSC), Precision, Recall, and Jaccard, which provide valuable quantitative insight on boundary overlap, precision, and quality of segmentation. Boundary accuracy was evaluated by the mean absolute distance (MAD), mean signed distance (MSD), and Hausdorff distance (HD), to quantify the comparative closeness and surface area.

To validate the segmentation uncertainty estimations, we apply thresholds. This involved normalizing the entropy estimates per slice ranged from 0 and 100, and varying the thresholds (T = 25, 50, 75) at different confidence levels as in [8]. Pixels exceeding the given threshold were labelled as uncertain and the remainder were cross-checked against the annotated mask to generate four disjoint classes: true positive (TP) (overlapping areas), false positive (FP) (over-prediction), false negative (FN) (under-prediction), and uncertain. As we lowered the uncertainty threshold, the FN and FP areas should have been filtered out while retaining the TP pixels. This validated that in areas where the model is confident, it is correct, while incorrect areas have high uncertainty. This permits clinical trust in areas of high model confidence. The entire spatial confidence map, along with the segmentation results, allows a framework for downstream clinical review.

## 3      Experiments and Results

### 3.1      Patient Data, Preprocessing and Implementation Details

Our patient dataset was collected after approval by the Western University Research Ethics Board using a GE Lightspeed 64-slice CT scanner and included 76 patients' cases, each containing 12 axial CT slices obtained post-ablation. All the images were in BMP format, grayscale, originally sized at 510 × 788 pixels, and were accompanied by manually annotated binary masks of the KAZ, which were generated by an expert. The 3D CT images were resliced radially around an approximate vertical axis of the KAZ every 15° into 2D CT images. This transformation ensured that the zone appears more consistently across 2D image samples. Each slice was resized to 256 × 256 pixels for computational feasibility to fit into the model, resulting in a dataset of 912 2D CT images, as shown in **Fig. 1**.

We normalized pixel intensities to a [0, 1] range, and split the dataset by patient into training (64%), validation (16%), and testing (20%) sets. This ensured that slices from the same patient did not appear in multiple subsets to avoid model bias. To enhance model reliability and reduce variance due to dataset partitioning, a nested 5-fold patient-wise cross-validation strategy was adopted. Each fold used a unique set of patients for training, validation, and testing, ensuring that no slices from a single patient were shared across splits.

On-the-fly data augmentation was applied to expand the appearance diversity while preserving label fidelity to compensate for the relatively small dataset. Augmentations were executed in TensorFlow eager mode, so a new stochastic version of every training image was generated for each epoch without materializing augmented files on disk. Each slice had a 30% chance of undergoing one or more spatial transformations: horizontal or vertical flip, translation of $\pm 10\,\%$ of the image extent, rotation of $\pm 20°$, or isotropic zoom between 0.9 and 1.1. Independently, there was a 30 % chance of a photometric adjustment that scales contrast between 0.8 and 1.2.

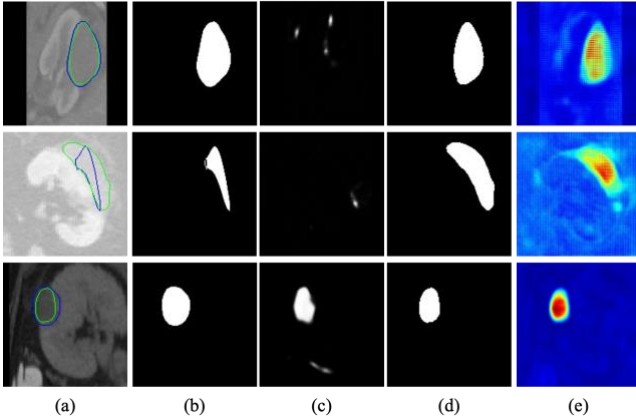

(a)        (b)        (c)        (d)        (e)

**Fig. 3.** XBoundNet++ results for three image slices from three different patients, in each row a) Original image, showing model prediction contour and clinical annotation, b) Clinically annotated mask, c) LeXNET++ prediction, d) XBoundNet++ prediction, e) The prediction attention heatmap from the convolutional layer before the sigmoid is applied. Attention shows higher gradient activation in red and thus more involvement in the resulting prediction, as it is more confident in the centre and is less confident at the boundaries.

### 3.2    Segmentation Results

**Fig. 3** shows the original image, mask, XBoundNet++ prediction, and the corresponding attention-based heatmap. These results show that the predictions generated by XBoundNet++ accurately align with the KAZ better than LeXNET++, as well as provide clarity on how strong the activations are that result in the arrival to the final prediction. This is evident in the first patient, where the KAZ, annotation mask, and model prediction all agree and cover the same area inside the kidney. While the second image may appear to be over-segmented, it is due to an incomplete annotation

mask. The model correctly delineated the full ablation zone, actually outperforming the human annotation. The third prediction correctly under-segments, as the manual annotation extends beyond the actual KAZ and kidney region.

**Table 1.** Ablation analysis on different metrics in XBoundNet++, with the cumulative addition (+) of new components in descending order, highlighting the best result in grey.

| Metrics / Models | Precision | Recall | DSC | Jaccard | MAD (pixels) | MSD (pixels) | HD (pixels) |
|---|---|---|---|---|---|---|---|
| LeXNet++ Baseline | 0.68±0.33 | 0.54±0.36 | 0.55±0.34 | 0.45±0.31 | 37.06±57.94 | 27.44±61.89 | 86.96±119.89 |
| +XBoundNet++ | 0.71±0.20 | 0.68±0.30 | 0.66±0.27 | 0.54±0.26 | 26.00±41.83 | 17.59±44.03 | 61.95±66.28 |
| +Augmentation | 0.82±0.18 | 0.80±0.23 | 0.78±0.19 | 0.76±0.17 | 15.48±26.10 | 8.19±27.22 | 43.47±55.10 |
| +Post-Processing | 0.81±0.18 | 0.83±0.19 | 0.81±0.17 | 0.71±0.18 | 12.47±29.55 | 6.28±30.76 | 28.29±37.16 |
| +CombinedLoss | 0.84±0.18 | 0.82±0.19 | 0.82±0.17 | 0.72±0.17 | 10.54±24.13 | 3.80±25.39 | 24.83±31.64 |
| +Ensemble | **0.88±0.11** | **0.83±0.13** | **0.84±0.10** | **0.74±0.13** | **6.89±4.33** | **-0.60±5.90** | **19.86±12.40** |

The results of the ablation study is shown in **Table 1** and was conducted to isolate the effect of each added XBoundNet++ component. Starting from the LeXNet++ [3] baseline, which lacks data augmentation, post-processing, and loss customization, we observe steady improvements across all metrics with each addition. XBoundNet++ alone improves DSC by 11%, recall by 12%, and HD by 25 pixels. Adding data augmentation further boosts DSC by 12%, Jaccard by 22%, and reduces MAD and MSD by over 9 pixels. Post-processing and the combined loss yield additional gains in boundary-related metrics, notably 3% DSC and a 15-pixel HD reduction. Finally, the ensemble improves all metrics, culminating in a 29% gain in DSC and Jaccard, 20% precision, and 67.1-pixel HD reduction compared to the baseline.

While these metrics clearly demonstrate the quality of our proposed model, it is important to consider that there is no clear ground-truth in this application because KAZ boundaries are inherently ambiguous and the manually-drawn masks are subject to user variability. Thus, quantitative gains do not always capture the full clinical value (i.e., rows 2 and 3 in **Fig. 3**, where the model outperformed the annotation – based on post-hoc review).

## 3.3    Model Transparency

As visualized in **Fig. 2**, early convolutional layers tend to activate broadly across the ablation zone, while deeper layers increasingly emphasize peripheral boundary regions, particularly near ambiguous areas. The heatmaps clearly illustrate a transition from low-level texture detection in early layers to high-level semantic abstraction in deeper layers, confirming that the network progressively refines its attention toward clinically relevant boundaries. As a result, we can clearly track information flow and decision making, revealing the model's focus, enabling trustworthy AI.

### 3.4    Uncertainty Analysis

**Fig. 4** illustrates a qualitative analysis of a given patients' KAZ region and provides more insight for clinicians. The prediction doesn't span over the healthy tissue at the bottom despite the manual annotation including it. The clinician can refer to the probability and uncertainty overlays to manually scrutinize areas with less confidence and higher uncertainty. The results show that decreasing the threshold leads to filtering out pixels of high uncertainty only.

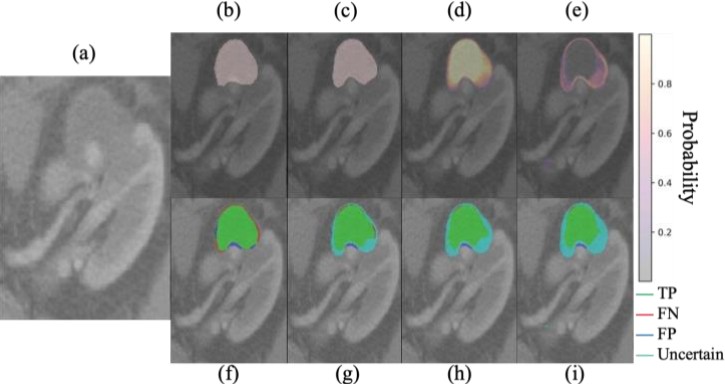

**Fig. 4.** XBoundNet++ results, uncertainty, probability, and thresholding visualized over a patient's CT slice. (a) CT original patient image slice, (b) Manually annotated mask, (c) XBoundNet++ predicted mask, (d) Probability map based on MC and ensembling, (e) Predicted entropy map from the probability map, (f) Uncertainty threshold = 100, (g) Uncertainty threshold = 75, (h) Uncertainty threshold = 50, (i) Uncertainty threshold = 25. It is desired that with more filtered out, more False Positives and False Negatives pixels are filtered out (marked uncertain), while True Positive pixels remain unfiltered.

## 4    Conclusions

In this work, we propose XBoundNet++, a novel deep learning segmentation framework that provides clinicians with several auxiliary interpretable and uncertainty tools to better equip them for clinically challenging contexts such as poor image contrast, no delineated boundary, or incomplete labels. The model excels at segmentation based on several key metrics, provides in-depth transparency using Grad-CAM, and uncertainty estimation generated by Bayesian MC dropout and model ensembling. By offering transparency, spatial uncertainty and probability overlays, XBoundNet++ enables more informed clinical review and supports safer, more trustworthy AI-assisted decision-making in interventional radiology.

   Future work can involve propagating the uncertainty the model's generated uncertainty into downstream clinical tasks or metrics, such as margin status, ablation volume, tumour volume changes, and residual tumour estimation.

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
