# OpenReview forum: "XBoundNet++: Uncertainty-Aware Segmentation of Kidney Ablation Zones"
_MICCAI.org/2025/Workshop/MSB_EMERGE — MSB EMERGE 2025 Conditionalrequiresmajorrevision_

### Official Review · Reviewer_eZMr · 2025-07-03

**Recommendation:** 3
**Confidence:** 4

**Clarity:**

The paper is generally clear but has some clarity issues that could be addressed with moderate revision

**Feedback:**

Title suggestion: Consider simplifying the title to “XBoundNet++: Uncertainty-Aware Segmentation of Kidney Ablation Zone” for clarity and focus.

On supervision and label quality: Given the authors' claim that obtaining accurate segmentation masks is particularly challenging, the use of a fully supervised approach with annotations from a single expert seems contradictory. Consider exploring or discussing alternatives, such as weakly supervised or unsupervised methods.

Label quality claim: The statement “Furthermore, it is well-established that only coarse and rough labels are provided by clinicians with the understanding that these are rough estimates that identify the area of interest.” is unsubstantiated. Please either provide references or revise the claim.

Literature claim: The statement “These contexts are not well studied in the literature and require particular care in terms of providing the clinician with model transparency and model uncertainty in order to trust the results and review the areas of relevance.” is misleading. There is significant existing work on model uncertainty and transparency in medical imaging. Relevant literature should be cited and discussed.

Missing discussion section: The paper lacks a Discussion section. This is important for addressing limitations (e.g., single annotator, low-resolution input, postprocessing reliance), interpreting results in context, and suggesting future directions (e.g., incorporating multi-rater labels or unsupervised training). A thoughtful discussion would help position the work within the broader literature.

Model description: The claim “XBoundNet++, an eXplainable Boundary-Aware modified ResU-Net++” lacks justification. The paper does not clearly demonstrate how the model is boundary-aware. Later statements such as “while deeper layers increasingly emphasize peripheral boundary regions” are speculative—deeper layers typically operate on high-level features and may not explicitly represent boundary information. Please clarify or rephrase.

Component sourcing: The sentence “Our architecture integrates components from LeXNet++ [3],[…] and several advanced architectures.” should clearly list which components are reused and which are novel contributions.

Postprocessing: “Next, we perform a morphological closing operation to seal small holes or gaps in the prediction if necessary.” Ideally, the model should produce clean outputs without requiring this. Please justify its use or evaluate its impact.

Connected components: The rule “instances where cysts are larger than the KAZ; in such a case, the second largest component is selected.” raises questions. How was this heuristic validated? Was manual verification done for all cases?

Image resizing: Downsampling from 510×788 to 256×256 may degrade segmentation accuracy. Consider a patch-based or multi-scale approach to preserve anatomical detail.

Ablation study: The ablation study is incomplete. Key components such as the ASPP bridge are not evaluated. Also missing are evaluations for baseline + augmentation, postprocessing, combined loss functions, and ensembling. A more systematic analysis would strengthen the experimental claims.

Feature maps and heatmaps: The statement “The heatmaps clearly illustrate a transition from low-level texture detection in early layers to high-level semantic abstraction in deeper layers, confirming that the network progressively refines its attention toward clinically relevant boundaries.” is speculative. There is no evidence provided that deeper layers specifically focus on clinically meaningful boundaries.

Figure quality:
    Figure 1: Please crop the images for improved clarity and focus.
    Figure 4: Provide higher-resolution images for better visual analysis.

**Justification:**

The paper tackles an important problem. However, the contribution is undermined by a lack of clarity what components belong to the proposed method, insufficient evaluation, missing related work, and unsupported claims.

**Reproducibility:**

Some amount of details available for reproducing the main results, and open access details are unclear

**Strengths:**

The paper addresses an important and clinically relevant application: accurate segmentation of ablation zones in kidney interventions.

**Summary:**

The authors present XBoundNet++, a novel network for the uncertainty-aware segmentation of Kidney Ablation Zones (KAZ). The method builds on LeXNet++ and uses a modified U-Net architecture with Monte Carlo (MC) dropout to produce probabilistic segmentation maps. The model is evaluated on a private dataset of 76 patients with expert-annotated segmentation masks and compared against LeXNet++ as a baseline.

**Weaknesses:**

Missing Related Work / Lack of SOTA Comparison: There is no comprehensive comparison to state-of-the-art (SOTA) methods. Several related works on uncertainty-aware segmentation, ablation zone analysis, and Bayesian deep learning are missing. The claim that the context is underexplored is inaccurate.

Supervision Assumptions: The authors stress the difficulty of obtaining accurate segmentations, yet proceed with a fully supervised approach based on annotations from a single expert. This contradiction is not sufficiently justified or at least discussed.

Model Description: The architectural details of XBoundNet++ are not well explained. The description of the model as “boundary-aware” is not substantiated as there is no explicit boundary loss or architectural component clearly linked to boundary modeling. Additionally, it is unclear what components are inherited or adapted from LeXNet++, and what constitutes a novel contribution.

Ablation Study: The ablation study is incomplete and omits key components like the ASPP bridge. The evaluation simply adds features without systematically isolating their effects. A proper study and comparison (e.g., baseline + augmentation + ASPP + loss + ensemble) is necessary to support the claims.

Experiments lack comparison to SOTA: The experimental design compares only to LeXNet++, which is insufficient. There is no comparison to nnU-Net, a de facto standard in medical image segmentation, nor to other uncertainty-aware segmentation approaches such as Probabilistic U-Net, Bayesian U-Net, or test-time augmentation methods. This makes it difficult to assess the true contribution of XBoundNet++ relative to existing methods in both segmentation performance and uncertainty quantification.

Unsupported Claims: Several statements throughout the manuscript are speculative or inaccurate, and are not supported by citations.

---

### Official Review · Reviewer_ZySX · 2025-07-08

**Recommendation:** 1
**Confidence:** 5

**Clarity:**

The paper is generally clear but has some clarity issues that could be addressed with moderate revision

**Feedback:**

Expand Dataset: i.e. Use public datasets like BRATS to increase sample size and diversity.

Clarify Trustworthy AI: Explicitly defined "trustworthy AI" but missing reference current standards (e.g., FUTURE AI BMJ 2024 paper).

Distinguish Explainability/Interpretability: There is a difference between interpretability (uncertainty estimation) and explainability (feature attribution). This is left ambiguous and should be used with more clarity.

Detailed Architecture Description: There is a board scope and very much does not get fully described in the following section. ⇾ low chance for reproducibility and lack of clarity.

Justify Bayesian Methods: Clearly explain the Bayesian components, underlying assumptions, motivation, and their impact on results.

Transparent Data Splitting: State rather absolute numbers for train/val/test splits instead of percentages, especially with small datasets.

Too much going on/ under-justified/evaluated techniques: Either thoroughly evaluate added techniques (e.g., Grad-CAM) or focus the scope.

Benchmark Against Relevant Methods: Compare with state-of-the-art uncertainty segmentation models and cite relevant literature. Here A list of interesting stuff for the authors:

    Kohl, Simon, et al. "A probabilistic u-net for segmentation of ambiguous images." Advances in neural information processing systems 31 (2018).

    Baumgartner, Christian F., et al. "Phiseg: Capturing uncertainty in medical image segmentation." Medical Image Computing and Computer Assisted Intervention–MICCAI 2019: 22nd International Conference, Shenzhen, China, October 13–17, 2019, Proceedings, Part II 22. Springer International Publishing, 2019.

    Fuchs, Moritz, Camila Gonzalez, and Anirban Mukhopadhyay. "Practical uncertainty quantification for brain tumor segmentation." Medical Imaging with Deep Learning. 2021.

    Chatterjee, Soumick, et al. "PULASki: Learning inter-rater variability using statistical distances to improve probabilistic segmentation." Medical image analysis (2025): 103623.,

    Monteiro, Miguel, et al. "Stochastic segmentation networks: Modelling spatially correlated aleatoric uncertainty." Advances in neural information processing systems 33 (2020): 12756-12767.

    https://link.springer.com/chapter/10.1007/978-3-031-16749-2_7)

Further, there are many more... Please do a proper literature review.

Specify Uncertainty Types: Explicitly state and analyze the types of uncertainty estimated.

Add Calibration Metrics: Report standard uncertainty calibration measures to quantify the value of uncertainty predictions.  i.e. calibration measure like MCE, Brier, NLL, GED or (Segmentation-based) ECE.

**Justification:**

The paper addresses a relevant clinical task and proposes a novel segmentation framework, the extremely small dataset, lack of methodological clarity, insufficient benchmarking, and limited evaluation of uncertainty significantly undermine its scientific value. Key details are missing, making the work difficult to reproduce or trust. These issues justify a strong reject, with encouragement to address these concerns in future revisions.

**Reproducibility:**

Some amount of details available for reproducing the main results, and open access details are unclear

**Strengths:**

Novelty: Proposes a boundary-aware, interpretable segmentation architecture tailored for ambiguous clinical labels.

Uncertainty Estimation: Integrates Bayesian Monte-Carlo dropout and ensembles for uncertainty quantification.

Interpretability: Provides layer-wise attention and Grad-CAM heatmaps to visualize model focus.

**Summary:**

The paper introduces XBoundNet++, a deep learning framework for kidney ablation zone segmentation in CT images, incorporating boundary-aware architecture, attention mechanisms, and Bayesian uncertainty estimation. Evaluated on a small dataset (76 patients, 912 slices), the model demonstrates strong segmentation performance and provides interpretability tools, but suffers from issues in reproducibility, dataset size, and lack of comparison to established uncertainty quantification methods.

**Weaknesses:**

In no particular order:

Extremely Small Dataset: Only 76 patients, which limits generalizability and statistical power.

Ambiguous Definition of Trustworthy AI: Fails to clearly define or reference standards for trustworthy AI as per recent literature.

Explainability vs. Interpretability: Conflates uncertainty estimation (interpretable) with true explainability; lacks feature-level traceability.

Unclear Architecture Description: The integration of multiple architectures is not sufficiently detailed for reproducibility.

Bayesian Approach Not Well Explained: The motivation and assumptions behind the "Bayesian" loss and inference are not clear.

Test Slice Generation: Insufficient detail on how test slices are generated and processed.

Non-standard Data Format for the dataset: Why using BMP instead of DICOM, omitting clinically relevant metadata that could be helpful to use and reproduce results.

Unclear Dataset Splitting: Does not provide absolute patient numbers for train/val/test splits, which is critical for such a small dataset as the numbers do not round well to absolute numbers.

Overly Ambitious Scope: Introduces techniques (like Grad-CAM) without thorough evaluation or justification.

Lack of Related Work Comparison: No comparison with relevant uncertainty segmentation methods (e.g., Probabilistic U-Net, PHISEG, VIMH, PULASki, SSN).

Uncertainty Type Not Specified: Does not clarify which uncertainty types are being estimated (aleatoric, epistemic, etc.).

No Uncertainty Calibration Metrics: Omits standard measures like MCE, Brier score, NLL, GED, or other Segmentation-based Calibration metric(SECE/USCE) to quantify uncertainty value.

---

### Official Review · Reviewer_9X9d · 2025-07-09

**Recommendation:** 2
**Confidence:** 3

**Clarity:**

The paper is generally clear but has some clarity issues that could be addressed with moderate revision

**Feedback:**

Please see suggestions in weaknesses section.

**Justification:**

I believe this paper addresses a relevant problem and presents promising results. However, in its current form, it lacks the clarity, contextual grounding, and experimental rigor needed to substantiate its claims. The absence of a related work section, limited dataset size, and insufficient benchmarking make it not yet suitable for publication.

**Reproducibility:**

Some amount of details available for reproducing the main results, and open access details are unclear

**Strengths:**

This study is driven by a clinically relevant motivation: it addresses the segmentation of KAZ, a clinically significant task in interventional oncology.

**Summary:**

This paper presents XBoundNet++, a deep learning framework for segmenting kidney ablation zones (KAZ) in post-treatment CT scans. The model integrates boundary-aware segmentation, layer-wise interpretability using a customized Grad-CAM, and uncertainty estimation through Bayesian Monte Carlo dropout, model ensembling, and normalized entropy. Compared to the baseline LeXNet++, XBoundNet++ demonstrates superior performance, as evaluated using several metrics: Precision, Recall, Dice Similarity Coefficient (DSC), Jaccard Index, Mean Absolute Distance, Mean Signed Distance, and Hausdorff Distance (refer to the last row of Table 1 for overall performance).  Additionally, an ablation study is conducted to assess the individual contribution of each component within the XBoundNet++ framework.

**Weaknesses:**

- The specific contributions and novelty of this paper are difficult to assess due to the absence of a dedicated related work section. A more comprehensive discussion is needed, including clear comparisons and appropriate references to existing studies.
- The model was trained on a relatively small dataset comprising only 76 patients. Evaluating the model on at least one external dataset is necessary to assess robustness of the results.
-Beyond LeXNet++, the paper does not include benchmarking against other widely adopted segmentation models such as 3D U-Net and nnU-Net. Including such comparisons is essential to properly contextualize the results within the current research domain.
- Analogously, the paper lacks a discussion of the limitations or potential challenges associated with the proposed framework. Including such a discussion would enhance the clarity, reproducibility, and real-world applicability of the work.

---

### Decision · Program_Chairs · 2025-07-18

**Decision:**

Conditional Accept (requires major revision)

**Comment:**

The paper is conditionally accepted to the EMERGE workshop. While the reviewers recognize the value of the contribution, the authors are required to address the following points in the camera-ready version:
1. Clearly explain why standard segmentation networks (e.g., U-Net, nnU-Net) were not considered for evaluation.
2. Include references to relevant prior work in the associated research areas to situate the contribution better.
3. Provide details regarding code and dataset availability to support reproducibility.

Acceptance is contingent on satisfactorily addressing these concerns.